# Exploring Platelet-Rich Plasma Therapy for Knee Osteoarthritis: An In-Depth Analysis

**DOI:** 10.3390/jfb15080221

**Published:** 2024-08-09

**Authors:** Florin Nicolae Blaga, Alexandru Stefan Nutiu, Alex Octavian Lupsa, Nicu Adrian Ghiurau, Silviu Valentin Vlad, Timea Claudia Ghitea

**Affiliations:** 1County Clinical Emergency Hospital of Oradea, 65 Gheorghe Doja Street, 410169 Oradea, Romania; blagaflorin86@yahoo.com (F.N.B.); alexandru_nutiu@yahoo.com (A.S.N.); lupsalex@icloud.com (A.O.L.); ghiurau.adrian@yahoo.com (N.A.G.); 2Department of Surgical Specialties, Faculty of Medicine and Pharmacy, University of Oradea, 10 1st Decembrie Street, 410073 Oradea, Romania; 3Pharmacy Department, Faculty of Medicine and Pharmacy, University of Oradea, 10 1st Decembrie Street, 410073 Oradea, Romania

**Keywords:** platelet-rich plasma, knee, osteoarthritis, treatment, orthopedic, intra-articular

## Abstract

The use of platelet-rich plasma (PRP) in all medical fields is currently gaining popularity (1). PRP is a biological product that can be defined as a segment of the plasma fraction of autologous blood with a platelet concentration level above the baseline (2). The fact that it has uses in tissue regeneration and wound healing has caught the eye of orthopedic surgeons as well, as intra-articular treatments have continued to evolve. Its benefits in the treatment of different osteoarticular pathologies are of great interest in the evolving orthopedic community, targeting mostly knee osteoarthritis, meniscus and ligament injuries (3). The purpose of this review is to update the reader on the current uses of platelet-rich plasma (PRP) in the treatment of knee osteoarthritis pathology and to provide clinical feedback on its uses in the fields of orthopedic and sports medicine practice (4). We proceeded in studying 180 titles and abstracts eligible for inclusion. Compared to alternative treatments, PRP injections greatly improve the function of the knee joint.

## 1. Introduction

Affecting over 300 million people, osteoarthritis is a degenerative joint disorder of the cartilage, targeting mostly women over 65 years, with 89% of the disorder site being the knee. Despite the increase in life expectancy, issues related to obesity are on the rise, along with an increasing incidence of sports-related injuries; consequently, the prevalence of knee pathologies continues to increase [1]. Hormonal imbalances [2], misuse of medications [3], inadequate rehabilitation following traumatic events or surgeries [4] and a lack of education about these conditions are all interconnected major factors [4]. 

Although widely studied and of great relevance in the modern medical world [5,6], the knee joint and the subsequent pathology that it can inherit still remain uncharted [7,8]. 

The presence of hyaluronic acid, along with its combined treatments involving extracorporeal shock wave therapy [9], has proven to be significantly more effective than monotherapy without hyaluronic acid [10].

Platelet-rich plasma (PRP) therapy has gained attention as a promising treatment for knee osteoarthritis (OA) due to its regenerative and anti-inflammatory properties. PRP is known for its ability to significantly reduce pain and improve knee function through the action of growth factors that promote tissue repair and cartilage regeneration [5,11]. Additionally, PRP’s anti-inflammatory cytokines help alleviate joint swelling and inflammation, enhancing overall symptom relief. As an autologous treatment, PRP also presents a minimally invasive alternative to surgical options, with a low risk of immune rejection or disease transmission [9,10].

Platelet-rich plasma (PRP) therapy has emerged as a noteworthy treatment for knee osteoarthritis (OA) due to its regenerative and anti-inflammatory effects [11]. PRP is effective for alleviating pain and enhancing knee function by leveraging growth factors that facilitate tissue repair and cartilage regeneration. It also helps reduce joint swelling and inflammation through its anti-inflammatory cytokines, offering significant symptom relief [12]. Additionally, as an autologous treatment, PRP is a minimally invasive option with a low risk of immune rejection or disease transmission, making it an appealing alternative to more invasive surgical procedures [13].

In addition to knee OA, PRP is used in various other medical areas. It aids in the treatment of chronic tendinitis, such as tennis elbow and Achilles tendinitis, by supporting tendon healing. In orthopedics, PRP promotes recovery from ligament injuries and enhances post-operative healing [14]. Its applications extend to dermatology as well, where it is utilized for skin rejuvenation and hair restoration, leveraging its regenerative properties to improve skin texture and stimulate hair growth. These diverse uses underscore PRP’s broad potential for supporting recovery and regeneration across different medical conditions [14].

The purpose of this review was to take into account the most common osteoarticular pathologies, of which we chose to tackle three frequent diseases and to detail the current use of platelet-rich plasma, the physiological pathways of its constituents and the benefits they garner.

## 2. Materials and Methods

### 2.1. Search Strategy

We conducted a review of the characteristics of platelet-rich plasma used as intra-articular treatment for knee pathologies, particularly focusing on knee osteoarthritis.

This narrative review was conducted using three databases: Google Scholar, PubMed and MDPI. In order to identify the most useful articles, the search strategy included combined keywords such as “knee osteoarthritis”, “PRP treatment”, “osteoarthritis”, “knee pathology” and “platelet-rich plasma.” The databases were searched for studies published between 1974 and 2023.

The search strategy was designed to identify studies on the anti-inflammatory, analgesic and anti-aging effects of platelet-rich plasma treatments on knee pathologies, mostly knee osteoarthritis.

### 2.2. Study Selection and Eligibility Criteria 

All electronically searched titles, selected abstracts and full-text publications were independently reviewed by a minimum of four reviewers. We included papers containing the keywords mentioned in our article. Although there was no restriction on the year of publication for the studies included, the majority of the papers were published after 2019. 

Through database research, 236 records were identified. After removing 20 duplicates, as well as 36 irrelevant articles, we proceeded to study the remaining 180 titles and abstracts eligible for inclusion. Finally, we included 105 references that met the criteria of the study.

#### Inclusion Criteria

Inclusion criteria for articles encompassed a diverse range of pathologies, including primary and secondary osteoarthritis of the knee, osteochondritis dissecans, degenerative meniscal lesions and meniscal and/or ligamentous injuries. Papers in which patients underwent various surgical procedures involving the knee, such as meniscectomy, meniscus-sparing surgery, meniscus replacement, ligament reconstruction and knee osteotomy, were also included in the review.

The focus of the review extended to the treatment of knee osteoarthritis, osteochondritis, meniscal and ligament injuries, as well as the evaluation of these treatments. The comprehensive review considered modern treatment methods, ranging from conservative and minimally invasive approaches to surgery, including the use of platelet-rich plasma, synovial mesenchymal stem cells, hyaluronic acid, extracorporeal shock wave therapy and high-intensity laser.

Exclusion criteria involved studies that did not meet the specified requirements, such as animal studies or those involving patients with rheumatoid arthritis. Additionally, papers published in languages other than English were excluded. Any disagreements regarding whether texts fit the inclusion or exclusion criteria were resolved through consensus. The search strategy is detailed in the flowchart shown in Figure 1.

## 3. Results

### Knee Osteoarthritis

Osteoarthritis (OA), a prevalent knee pathology, affects approximately 300 million people globally, with increasing incidence in individuals over 65 years [15,16]. Knee OA, representing 89% of the global osteoarthritis segment, is particularly burdensome to the healthcare system [17]. Factors contributing to OA include age, obesity, genetic factors and mechanical stress [18,19]. The degenerative process occurs in three stages, involving proteolytic degradation, fibrillation and degradation of cartilage [20] (Figure 2).

The third phase includes collagen and proteoglycan degradation, increased pro-inflammatory enzymes and inhibited synthesis of matrix constituents [19,20,21,22,23,24,25] (Figure 3).

Age is a major contributor to OA, affecting the structural properties of cartilage’s extracellular matrix [26]. OA leads to degeneration in articular cartilage, subchondral bone, connective tissue and abnormal joint metabolism [27,28]. Structural and molecular changes in the entire joint characterize OA [29]. Cartilage erosion, tissue loss, sub-chondral bone cyst formation and osteophytes are common [30]. Knee OA symptoms include pain, disability, stiffness, loss of function and swelling, as diagnosed via radiography, MRI or CT scan [31] (Figure 4).

Degeneration of cartilage integrity and chondrocyte function alteration are major factors in OA knees. The cartilage tends to erode, and significant loss of tissue can appear due to the hyaline cartilage accumulating water, the reduction in proteoglycan levels and reduced cartilage stiffness. Formation of subchondral bone cysts are common, as well as osteophytes [30].

The main symptoms of knee osteoarthritis are pain, disability, stiffness, loss of function and swelling; confirmation of the diagnosis is performed via knee radiography, MRI or CT scan [31].

Ligament and meniscus injuries are prevalent in younger patients (25–50 years old), increasing the risk of OA in the long term [32,33]. Contrary to previous beliefs, these injuries may not always result from traumatic events, suggesting a connection with OA development [34,35,36]. Strategies to prevent and delay OA onset are crucial, especially in this demographic [37].

## 4. Pharmacological Treatment of Knee Osteoarthritis

Knee osteoarticular (KOA) pathology is often treated from a short-term perspective, but considering patients’ long-term experiences with KOA, it is crucial to emphasize long-term approaches [38,39]. Despite being cost-effective and simple, patient-controlled measures such as lifestyle changes and weight reduction are frequently overlooked [40,41]. Initial treatment for KOA aims for symptomatic relief [42], employing non-pharmacological and pharmacological methods. However, these approaches may offer short-lived benefits [43], with corticosteroid injections potentially worsening symptoms through cartilage degeneration [44,45]. The benefits of these kinds of treatments are usually short lived and the side effects that accompany especially injections with corticosteroids (cartilage degeneration) may further the symptoms even more [46].

### 4.1. Platelet Rich Plasma

Platelets, or thrombocytes as they are also called, emerge from the bone marrow. Platelets appear as nucleated, discoid cellular elements of different sizes and, remarkably, are the least dense of all blood cells, at about 2 μm in diameter. The normal number of platelets circulating in the blood stream of healthy individuals varies from 150,000 to 400,000 platelets per μL [47,48,49]. The primary process of the platelets is aggregation. Their role in hemostasis is ensured through 3 functions: adhesion, activation and aggregation. Vascular injuries initiate platelet activation, releasing specific factors that ensure coagulation through the secretory granules. Each platelet contains about 50–80 granules. There are three types of known secretory granules: dense granules, o-granules and lysosomes [50].

Although the initial understanding of platelets was that their role was solely that of a hemostatic agent, new research and advances in technology have resulted in their reevaluation. Studies have suggested that platelets have a role in inflammation, stem cell migration and proliferation and angiogenesis through the growth factors and cytokines that they contain [51].

When the platelets in PRP are activated, the P-granules go through degranulation and produce cytokines and growth factors that flow in the surrounding cellular space. For the use of PRP in common practice, the growth factors that are emitted and interest us most today are the vascular endothelial growth factor, transformative growth factor beta (TGF-β1), platelet-derived epidermal growth factor (PDEGF), insulin-like growth factor (IGF) and basic fibroblast growth factor (b-FGF or FGF-2). We highlight each of these growth factors that display several specific traits in Table 1 [32]. What is common among all and of great interest to us is their angiogenesis stimulation capacity [49,52,53].

### 4.2. Platelet Rich Plasma Preparation

PRP, which is obtained through differential centrifugation, can be prepared in two ways: open (risking contamination) [59,60,61] and closed (using anticoagulants) [62]. Centrifugation yields a three-layered display, with ongoing debate about adding leukocytes to PRP [63,64,65]. Activation debates persist, with substances such as calcium gluconate initiating clotting for localized effects [66,67].

Optimal PRP concentration has been debated; current devices achieve 2–5 times baseline levels, but levels above 2.5 times may inhibit the desired action [68,69,70]. Growth factors released post-administration last up to a year, prompting multiple close-interval administrations due to platelet half-life [64].

Studies have explored carriers such as gelatin hydrogel, hydroxyapatite and chitosan PRP hybrids for enhanced growth factor efficiency and prolonged effects, showing promise in animal tests [71,72].

## 5. Benefits of PRP Treatment in Knee Pathology

The interest of orthopedic and sports medicine practices regarding PRP treatments for use in treating ligament, tendon and bone lesions is surging [73]. The growth factors dispersed by PRP play a major role in promoting cell recruitment, angiogenesis and proliferation, resulting in a reduction in inflammatory enzymes [73,74]. 

PRP has a role in improving the metabolic functions of injured structures by transmitting a regenerative signal that could affect the proliferation of stem cells, with a positive effect on chondrogenesis [61,75,76]. 

The growth factors contained in platelets contribute to cartilage proliferation, and when applied to chondrocytes, they tend to help with protein transcription, cell growth, cell migration and matrix synthesis as a whole. They signal the regenerative cascade and tissue healing, while containing the inflammatory response [77]. In joints that suffer from osteoarthritis, PRP affects local and infiltrating cells, synovial and endothelial cells, cartilage and bone cellular elements [78]. It may halt the advance of joint disease by mitigating inflammation and angiogenetic processes, while decreasing cartilage catabolism and increasing anabolism [51]. 

All of these successful PRP uses could easily imply that it has a use as a primary analgesic treatment by accelerating the proliferation of tenocytes, osteoblasts and mesenchymal stem cells [79,80]. By involving the whole joint complex, through the stimulation of cartilage proliferation, PRP injections can achieve clinical improvement and short-term remission of OA symptoms, even delaying the need for knee arthroplasty [81]. Although study results still are inconclusive, PRP treatments yield better results than hyaluronic acid or a placebo in all stages of knee OA [82]. Compared to HA injections, PRP offers more advantages in the conservative treatment of OA, such as better joint function and better long-term amelioration of symptoms; all patients have superior outcomes at their 3, 6 and 12 month evaluations when compared to injections with placebo, steroids or HA [83,84,85]. WOMAC scores are also lower (favorable) in all cases of PRP use when compared to HA or corticosteroid treatment [86]. 

Studies mostly tend to favor PRP treatment, as it is overall safe and delivers good outcomes, although it does have several disadvantages [32] (Table 2).

Jang et al. conducted a review of 65 patients with knee osteoarthritis treated with a single intra-articular PRP administration. Clinical improvement was reported by most patients at 6 months, but the effects diminished one year after the injection [87]. In another study on 30 patients with knee chondropathy (Outerbridge I to III), Torrero et al. observed positive outcomes with a single intra-articular PRP administration, showing clinical improvement at the 6-month mark [88].

Hart et al. conducted a trial on 51 patients with chondromalacia grades II and III, spanning one year with nine autologous PRP injections, demonstrating significant clinical improvement [89]. In a randomized controlled trial by Patel et al. on bilateral osteoarthritis with 78 patients (156 knees), those receiving PRP injections (either a single dose or two injections) showed better results than the group receiving saline injections. The single dose with filtered concentrated white blood cells proved as effective as two PRP injections in terms of clinical improvement [90].

Filardo et al. compared two PRP preparation methods in 144 patients with osteoarthritis and degenerative cartilage lesions, finding significant clinical improvement in both groups at the one-year mark. The single-spinning method led to less ache and edema reported by patients [81].

Bansal et al. selected 150 randomized patients over three years, treating 75 with PRP and 75 with HA. Both groups showed significant improvement at one month, but the PRP-treated group retained benefits even at one year, while the HA-treated group showed a decline at subsequent controls (3, 6, 9 and 12 months) [69].

In total knee arthroplasty (TKA), the use of PRP has been documented in publications based on its effects, as shown in Table 3.

## 6. Discussion

Clinical studies provide compelling evidence suggesting the potential efficacy of PRP treatments for addressing osteoarticular degenerative joint pathology of the knee [31]. In comparison to alternative interventions, according to a 2017 study, PRP injections demonstrated notable improvement in knee joint function and served as a superior analgesic option [97]. The ease of preparation and administration, coupled with minimal need for extensive medical devices, positions PRP as a practical choice for regular use in orthopedic and sports medicine practices [98].

Several key mechanisms contribute to PRP therapy effectiveness in treating knee OA [32]. One of the primary mechanisms is the modulation of inflammation. PRP contains anti-inflammatory cytokines, such as interleukin-1 receptor antagonist (IL-1ra) and soluble tumor necrosis factor receptors, which inhibit pro-inflammatory cytokines, such as interleukin-1 (IL-1) and tumor necrosis factor-alpha (TNF-α) [99]. This action reduces joint inflammation, swelling and pain, improving overall symptoms [100].

Another significant mechanism is tissue regeneration. PRP is rich in growth factors, including platelet-derived growth factor (PDGF), transforming growth factor-beta (TGF-β) and vascular endothelial growth factor (VEGF). Another significant mechanism is tissue regeneration. PRP is rich in growth factors, including platelet-derived growth factor (PDGF), transforming growth factor-beta (TGF-β) and vascular endothelial growth factor (VEGF) [101]. These growth factors stimulate cellular proliferation, differentiation and angiogenesis, which are crucial for repairing damaged tissues. TGF-β, for instance, promotes chondrocyte proliferation and extracellular matrix production, aiding in cartilage repair. PDGF and VEGF enhance the formation of new blood vessels, improving nutrient and oxygen delivery to the damaged areas [102].

Additionally, PRP supports cartilage repair by influencing both anabolic and catabolic processes. It increases the production of cartilage matrix components, such as collagen type II and aggrecan, while decreasing the levels of matrix metalloproteinases (MMPs) that break down cartilage. This shift towards anabolic activity helps maintain and potentially restore cartilage integrity, which is essential for joint function [103].

Clinical evidence supports these mechanisms, with studies indicating significant improvements in pain and joint function following PRP treatment. For example, research by Smith et al. (2019) showed that PRP treatment led to meaningful improvements in clinical outcomes compared to other treatments, such as a placebo or hyaluronic acid injections, highlighting the effectiveness of PRP’s biological action for managing knee OA [31].

However, according to Tey in 2022, the variability in PRP preparation modes, influenced by physician preference, poses a challenge in establishing universal guidelines applicable to the majority of patients [104]. Further research is essential in order to refine and define specific protocols that account for the diverse ways in which PRP is prepared. While ongoing studies show promise, the absence of standardized guidelines underscores the need for additional investigation to establish comprehensive and universally applicable protocols [105].

Despite its advantages, PRP treatment can cause several adverse effects. Common local reactions include pain, swelling and irritation at the injection site, which are generally mild and temporary. In rare cases, infections may occur if aseptic techniques are not strictly followed [106]. Some patients may experience a post-injection flare, marked by a temporary increase in pain and inflammation. Additionally, the lack of standardization in PRP preparation can lead to inconsistent outcomes, making the treatment’s efficacy variable. On the other hand, PRP therapy offers promising benefits for knee osteoarthritis, including significant pain relief and improved joint function [107]. The growth factors in PRP, such as TGF-β and VEGF, promote tissue repair and cartilage regeneration. Additionally, PRP’s anti-inflammatory properties, mediated by cytokines such as IL-1ra, help reduce synovitis and joint swelling [108]. As a minimally invasive treatment, PRP poses a low risk of immune rejection or disease transmission since it uses the patient’s own blood components [60].

The adaptability of PRP treatments to diverse clinical settings, coupled with its relative simplicity and safety profile, supports its potential as a valuable therapeutic option for managing knee osteoarthritis pathology [109,110]. Although precise guidelines are currently elusive, the encouraging results from recent studies contribute to the evolving understanding of PRP’s role in knee joint health [111]. As research progresses, it is anticipated that a more nuanced and detailed framework for PRP application will emerge, offering clinicians clearer insights into its optimal use for improved patient outcomes.

PRP therapy offers regenerative potential and anti-inflammatory benefits by utilizing the patient’s own blood components, making it a minimally invasive option with a low risk of immune reaction [112]. However, the variability in PRP preparation and uncertain long-term safety are notable drawbacks. By contrast, corticosteroid injections provide quick and effective pain relief and inflammation reduction, but their benefits are typically short-lived, and repeated use can lead to cartilage damage and other side effects [113].

Hyaluronic acid injections offer longer-lasting relief by improving joint lubrication but lack regenerative benefits and can be costly. Physical therapy is a non-invasive approach that focuses on muscle strengthening and joint stability, promoting long-term joint health [114]. However, its effectiveness depends on patient compliance and consistent participation. Surgical interventions such as arthroscopy and total knee replacement provide definitive treatment, particularly for severe cases, but are invasive and come with significant risks, including infection and lengthy recovery times [115].

PRP therapy typically costs between $500 and $2000 per injection, making it more expensive than corticosteroid ($100–$500) and hyaluronic acid ($300–$800) injections. Despite the higher cost, PRP offers potential long-term benefits through its regenerative effects, which can lead to sustained improvements in pain and function [116].

Corticosteroid injections provide quick, short-term relief but may require frequent administration due to diminishing efficacy and potential side effects. Hyaluronic acid injections offer longer-lasting relief compared to corticosteroids but lack the regenerative benefits of PRP [117].

Physical therapy, costing $50–150 per session, is cost-effective for long-term joint health but requires patient commitment and time [118]. Surgical interventions are significantly more expensive, ranging from $15,000 to $50,000, and involve higher risks and recovery times. PRP may be a cost-effective option for patients seeking to delay or avoid surgery while potentially achieving longer-term relief [119].

The evolving understanding of knee pathologies, specifically the intricate interplay between aging, trauma and the onset of osteoarthritis (OA), has stimulated the exploration of diverse preventive strategies and prompted a reevaluation of conventional perspectives on causative factors. Continuous investigation and refinement of these concepts are imperative for the development of effective interventions using platelet-rich plasma (PRP) to alleviate the burden of OA on individuals and healthcare systems. Recent references emphasizing these aspects have become crucial for advancing our understanding and refining treatment approaches, laying the foundation for the utilization of PRP as a promising intervention for managing knee pathologies. This ongoing exploration is essential for tailoring PRP applications to address the unique factors contributing to knee pathologies and offering targeted and innovative solutions for improved patient outcomes.

## 7. Conclusions

Clinical studies have yielded enough evidence to suggest that PRP treatments could be helpful for the osteoarticular degenerative joint pathology of the knee. Compared to alternative treatments, PRP injections greatly improve the function of the knee joint and exhibit improved analgesic properties.

The fact that preparation and administration of PRP is relatively easy, safe and does not require a vast array of medical devices makes it a good option for everyday use in orthopedic or sports medicine practices.

Considering the way that each preparation mode of PRP varies and depending on the physicians’ preference, it is hard to choose an exact guideline that would apply to the vast majority of patients. More research has to be conducted in order to establish exact guidelines, but incoming studies and results are promising.

## Figures and Tables

**Figure 1 jfb-15-00221-f001:**
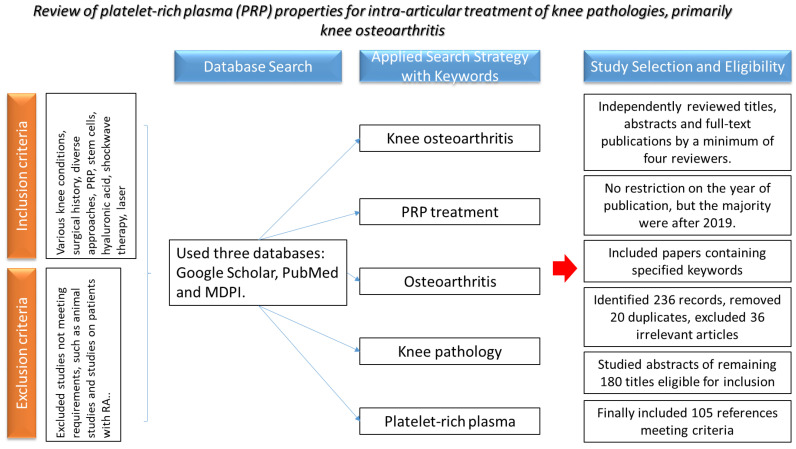
Flow chart.

**Figure 2 jfb-15-00221-f002:**
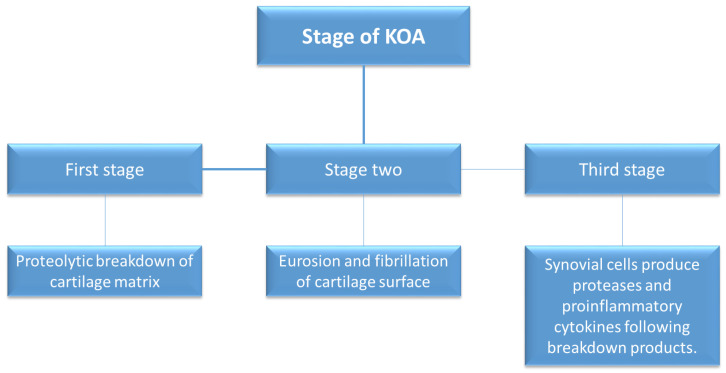
Stages of KOA.

**Figure 3 jfb-15-00221-f003:**
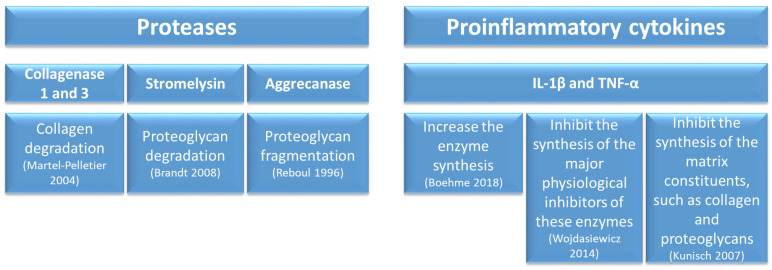
Third-stage KOA physipathological pathways [19,20,21,22,23,24,25].

**Figure 4 jfb-15-00221-f004:**
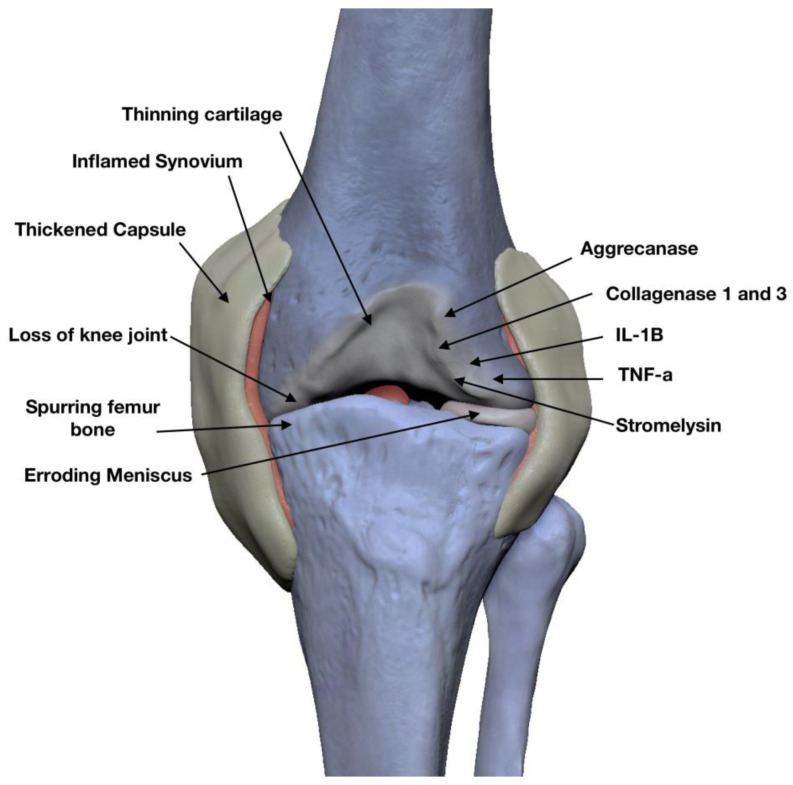
Structural and molecular changes in KOA. Adapted with permission of Bogdan-Viorel Leahu.

**Table 1 jfb-15-00221-t001:** Most-studied growth factors and specific traits.

Function	Growth Factors	Ref.
Activates the production of KGF. Regulates angiogenesis and wound contraction. Promotes collagen synthesis, matrix and epithelialization. Is responsible for the growth and differentiation of fibroblasts, myoblasts, osteoblasts, nerve cells, endothelial cells, keratinocytes and chondrocytes. Acts as a mitogen for mesenchymal stem cells. Stimulates the proliferation of myoblasts.	Basic Fibroblast Growth Factor (b-FGF)	[54,55,56,57,58,59]
Induces neovascularization by promoting proliferation and migration of macrovascular endothelial cells. Promotes angiogenesis and participates in the formation of blood vessel lumen indirectly through the release of nitric oxide. Initiates the regeneration of blood circulation and supports wound healing. Activates the synthesis of metalloproteinase and is involved in the degradation of interstitial collagen types 1, 2 and 3. Stimulates the chemotaxis of macrophages and neutrophils.	Vascular Endothelial Growth Factor (VEGF/VEP)	[54,55,56,57,58,60]
Stimulates endothelial angiogenesis. Regulates the secretion of collagenase. Stimulates epithelial and mesenchymal mitogenesis. Supports wound healing by stimulating the proliferation of keratinocytes and dermal fibroblasts.	Platelet-Derived Epidermal Growth Factor (PDEGF)	[55,56,57,60]
Stimulates endothelial chemotaxis and angiogenesis. Participates in the regulation of the balance between fibrosis and myocyte regeneration. Inhibits the formation of osteoclasts and bone resorption. Promotes chondrocyte proliferation and extracellular matrix synthesis, essential for cartilage repair. Inhibits the proliferation of macrophages and lymphocytes. Stimulates the chemotaxis of fibroblasts. Increases the synthesis of type I collagen and fibronectin and regulates the secretion of collagenase. Stimulates or inhibits endothelial, fibroblastic and osteoblastic mitogenesis. Inhibits DNA synthesis in human fibroblasts. Regulates the mitogenic action of other growth factors.	Transformative Growth Factor Beta (TGF-β1)	[54,55,57,58,60,61]
Stimulates the growth of myoblasts and fibroblasts. Activates the synthesis of collagenase and prostaglandin E2 in fibroblasts. Regulates the metabolism of articular cartilage through increased synthesis of collagen and matrix osteon. Stimulates cartilage growth, bone matrix formation and replication of preosteoblasts and osteoblasts. Together with PDGF, it can increase the speed and quality of wound healing by activating collagen synthesis. Mediates the growth and repair of skeletal muscles.	Insulin-like Growth Factor (IGF)	[55,56,57,58,59,60]

**Table 2 jfb-15-00221-t002:** Advantages and disadvantages of PRP treatment in the most recent studies.

Criteria	Benefits	Challenges	Other Considerations	Ref.
Minimal Invasiveness	✓	✓	Does not involve any surgery, incisions or healing	[84]
Rapid Preparation	✓	-	Does not require any preservative	[84]
Compatibility with Patient Cells	✓	-	Use of patient cells without any further modification	[85]
Comprehensive Therapeutic Effects	✓	-	Can simultaneously reduce synovial inflammation, protect cartilage and reduce pain	[86]
Contaminant Reduction	✓	-	Minimization of blood-borne contaminants	[85]
Accelerated Recovery Time	✓	-	Recovery period reduced	[84]
Enhanced Biocompatibility	✓	-	Does not elicit an immune response	[84]
Morbidity at Injection Site	-	✓	Disadvantage only at the local level	[86]
Standardization of Methods	-	✓	Does not exist	[86]
Scar Tissue and Calcification	-	✓	Local risk	[85]
Optimal Processing and Concentration	-	✓	Incompletely elucidated	[85]
Risk of Infections	-	✓	Disadvantage only at the local level	[86]
Risk of Allergic Reactions	-	✓	Disadvantage only at the local level	[84]
Unknown Frequency and Volume	-	✓	Does not exist	[85]
Contraindications for Certain Conditions	-	✓	Incompletely known	[86]

✓ = presence of benefits/challenges, - = absence of benefits/challenges

**Table 3 jfb-15-00221-t003:** Advantages and disadvantages of PRP treatment in TKA.

Effects	Without PRP	With PRP	Ref.
Post-operative verbal pain scale		✓	[91]
Increased success rates		✓	[92]
Reduced blood loss	✓		
Improved wound healing rate		✓	[93]
Better control of post-operative pain		✓	[94]
Knee range of motion	✓	-	[91]
Manipulation rates up to 3 months post-operative	✓	-	[95]
The circumference of the operated joint		✓	[96]

PRP = Platelet-rich plasma, ✓ = presence of benefits/challenges, - = absence of benefits/challenges.

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
