# Peer review of "Exploring Platelet-Rich Plasma Therapy for Knee Osteoarthritis: An In-Depth Analysis"

_jfb, 2024, doi:10.3390/jfb15080221_

Round 1

Reviewer 1 Report

Comments and Suggestions for Authors

This review focuses on platelet-rich plasma therapy for knee osteoarthritis. The paper presents the latest research and provides some results on the possibilities of platelet-rich plasma application. The Introduction section describes the recent challenges and the Materials and Methods section describes the  eligibility criteria and inclusion criteria.The following sections provide sufficient information. The discussion and conclusion confirmed the data presented.

The illustration of platelet morphology as well as its activation may provide more comprehensive information to the reader.

In this way, the paper presented could be published after minor revision.

Author Response

Reviewer 1

Dear Reviewer,

Thank you very much for accepting our manuscript for publication. We greatly appreciate your thorough review and valuable feedback, which have significantly improved the quality of our work. Your time and effort in evaluating our study are deeply appreciated. We are excited to see our research published and are grateful for your contribution to this achievement.

Sincerely,

The authors!

Reviewer 2 Report

Comments and Suggestions for Authors

The enclosed review of PRP as a treatment for osteoarthritis of the knee discusses recent literature in the field and focuses on the potential benefits that PRP offers for this type of injury.  As PRP-based therapy for joint lesions has observed wide spread application in both human and veterinary medical care for decades it is important to note that this constitutes a broad ranged topic.  This being strongly associated with the variability of what a given PRP treatment might entail preparation wise from clinic to clinic, which is touched on by the authors.  Therefore the limiting of scope to a single regional application and centering on potential modes of action described in literature appears to be an effective way of maintaining a focused dialogue.  The highlighting of impact for different growth factors as a result of PRP application is also an important aspect to detail and the inclusion of a table indicating references associated with each growth factor is a good addition.  Similarly the highlighting of different preparation methods, including initial activation or not pre-application (covered in lines 164-165), offer insight into the on-going discussions surrounding the optimal application of this biologic treatment.  While the methods, results, and discussion appeared well-formulated, the introduction section would strongly benefit from additional content to establish the background of PRP as this would provide readers for a better lead in to the rest of the review's material.  There appears to be this kind of information actually later on in the results section but it is the opinion of this reviewer that this kind of information would be better served prior to the methods so as to establish the focus of the review as a whole.  Additionally as a more minor comment, there appear to be multiple cases in which the text in section is hyphenated randomly and may be a formatting issue as a result of using one of the MDPI template.  A fairly minor note but did appear in several instances such as line 51.  Based on these comments it is the recommendation of this reviewer that the enclosed work be considered ready for acceptance after minor revision to the introduction section to better establish the focus and scope of the review.

Author Response

Reviewer 2

Firstly, we, the authors of the present manuscript wish to thank you for thoughtful commentary you have provided to improve the quality of the paper. We are very grateful for the time and effort you have devoted to this task. We have extensively revised my manuscript according to the recommendations. All changes in the text and the new figures that we have redesigned are highlighted. Please, see the point-by-point answers to your comments below. All correction was highlighted in the manuscript.

  1. The enclosed review of PRP as a treatment for osteoarthritis of the knee discusses recent literature in the field and focuses on the potential benefits that PRP offers for this type of injury. As PRP-based therapy for joint lesions has observed wide spread application in both human and veterinary medical care for decades it is important to note that this constitutes a broad ranged topic.  This being strongly associated with the variability of what a given PRP treatment might entail preparation wise from clinic to clinic, which is touched on by the authors.  Therefore the limiting of scope to a single regional application and centering on potential modes of action described in literature appears to be an effective way of maintaining a focused dialogue.  The highlighting of impact for different growth factors as a result of PRP application is also an important aspect to detail and the inclusion of a table indicating references associated with each growth factor is a good addition. 

Response 1: Thank you very much for suggestion. We, the authors, have completed the table 1.

  1. Similarly the highlighting of different preparation methods, including initial activation or not pre-application (covered in lines 164-165), offer insight into the on-going discussions surrounding the optimal application of this biologic treatment. While the methods, results, and discussion appeared well-formulated, the introduction section would strongly benefit from additional content to establish the background of PRP as this would provide readers for a better lead in to the rest of the review's material.  There appears to be this kind of information actually later on in the results section but it is the opinion of this reviewer that this kind of information would be better served prior to the methods so as to establish the focus of the review as a whole. 

Response 2: Thank you for the comment. We have added in introduction the benefits of PRP. (lines 40-63)

  1. Additionally as a more minor comment, there appear to be multiple cases in which the text in section is hyphenated randomly and may be a formatting issue as a result of using one of the MDPI template. A fairly minor note but did appear in several instances such as line 51.  Based on these comments it is the recommendation of this reviewer that the enclosed work be considered ready for acceptance after minor revision to the introduction section to better establish the focus and scope of the review.

Response 3: Thank you very much for observation. We have corrected all of the formating mistakes.

Reviewer 3 Report

Comments and Suggestions for Authors

Platelet-rich plasma (PRP) therapy for knee osteoarthritis represents a significant advancement in orthopedic and sports medicine. In this manuscript the author highlights PRP's efficacy in improving knee joint function and providing pain relief, surpassing traditional treatments such as hyaluronic acid and corticosteroids. PRP's minimal invasiveness, safety, and ease of preparation make it an attractive option for clinical practice. As the understanding of PRP's biological mechanisms and clinical applications continues to evolve, it holds promise as a cornerstone treatment for knee osteoarthritis, potentially delaying the need for surgical interventions and improving the quality of life for patients. This paper is well organized and can be published 

 with a minor revision. Below are my concerns:

1. Include a detailed discussion of the adverse effects of PRP treatment. Ensure that both benefits and risks are adequately addressed.

2. Provide a thorough comparative analysis with other treatment modalities. Discuss the relative advantages and disadvantages of PRP compared to other treatments.

3. Explain the mechanisms of action of PRP in more detail. Ensure that the explanation is supported by evidence from the included studies.

4. Discuss the cost-effectiveness of PRP treatment. Compare the costs and benefits with other treatment options.

Comments on the Quality of English Language

Minor editing of English language required

Author Response

Reviewer 3

Firstly, we, the authors of the present manuscript wish to thank you for thoughtful commentary you have provided to improve the quality of the paper. We are very grateful for the time and effort you have devoted to this task. We have extensively revised my manuscript according to the recommendations. All changes in the text and the new figures that we have redesigned are highlighted. Please, see the point-by-point answers to your comments below. All correction was highlighted in the manuscript.

Platelet-rich plasma (PRP) therapy for knee osteoarthritis represents a significant advancement in orthopedic and sports medicine. In this manuscript the author highlights PRP's efficacy in improving knee joint function and providing pain relief, surpassing traditional treatments such as hyaluronic acid and corticosteroids. PRP's minimal invasiveness, safety, and ease of preparation make it an attractive option for clinical practice. As the understanding of PRP's biological mechanisms and clinical applications continues to evolve, it holds promise as a cornerstone treatment for knee osteoarthritis, potentially delaying the need for surgical interventions and improving the quality of life for patients. This paper is well organized and can be published with a minor revision. Below are my concerns:

  1. Include a detailed discussion of the adverse effects of PRP treatment. Ensure that both benefits and risks are adequately addressed.

Response 1. Thank you for the observation. We, the authors, have completed the discussion (lines 271-283).

  1. Provide a thorough comparative analysis with other treatment modalities. Discuss the relative advantages and disadvantages of PRP compared to other treatments.

Response 2. Thank you for the suggestion. We, the authors, have completed the discussion (lines 291-303).

  1. Explain the mechanisms of action of PRP in more detail. Ensure that the explanation is supported by evidence from the included studies.

Response 3. Thank you for the comment. We, the authors, have completed the discussion (lines 238-264).

  1. Discuss the cost-effectiveness of PRP treatment. Compare the costs and benefits with other treatment options.

Response 4. Thank you for the observation. We, the authors, have completed the discussion (lines 304-316).
